# Discriminator-Actor-Critic: Addressing Sample Inefficiency and Reward Bias in Adversarial Imitation Learning

**Ilya Kostrikov**[1,2,*], **Kumar Krishna Agrawal**[2,†], **Debidatta Dwibedi**[2,†], **Sergey Levine**[2], and **Jonathan Tompson**[2]

[1]Courant Institute of Mathematical Sciences, New York University, New York, NY
[2]Google Brain, Mountain View, CA

## ABSTRACT

Algorithms for imitation learning based on adversarial optimization, such as generative adversarial imitation learning (GAIL) and adversarial inverse reinforcement learning (AIRL), can effectively mimic demonstrated behaviours by employing both reward and reinforcement learning (RL). However, applications of such algorithms are challenged by the inherent instability and poor sample efficiency of on-policy RL. In particular, the inadequate handling of absorbing states in canonical implementations of RL environments causes an implicit bias in reward functions used by these algorithms. While these biases might work well for some environments, they lead to sub-optimal behaviors in others. Moreover, despite the ability of these algorithms to learn from a few demonstrations, they require a prohibitively large number of the environment interactions for many real-world applications. To address these issues, we first propose to extend the environment MDP with absorbing states which leads to task-independent, and more importantly, unbiased rewards. Secondly, we introduce an off-policy learning algorithm, which we refer to as Discriminator-Actor-Critic. We demonstrate the effectiveness of proper handling of absorbing states, while empirically improving the sample efficiency by an average factor of 10. Our implementation is available online [1].

## 1 INTRODUCTION

The Adversarial Imitation Learning (AIL) class of algorithms learns a policy that robustly imitates an expert's actions via a collection of expert demonstrations, an adversarial discriminator and a reinforcement learning method. For example, the Generative Adversarial Imitation Learning (GAIL) algorithm (Ho & Ermon, 2016) uses a discriminator reward and a policy gradient algorithm to imitate an expert RL policy. Similarly, the Adversarial Inverse Reinforcement Learning (AIRL) algorithm (Fu et al., 2017) makes use of a modified GAIL discriminator to recover a reward function to perform Inverse Reinforcement Learning (IRL) (Abbeel & Ng, 2004). Additionally, this subsequent dense reward is robust to changes in dynamics or environment properties. Importantly, AIL algorithms such as GAIL and AIRL, obtain higher performance than supervised Behavioral Cloning (BC) when using a small number of expert demonstrations; experimentally suggesting that AIL algorithms alleviate some of the distributional drift (Ross et al., 2011) issues associated with BC. However, these AIL methods suffer from two important issues that will be addressed by this work: 1) a large number of policy interactions with the learning environment is required for policy convergence and 2) although in principle these methods can learn rewards for absorbing states, the original implementations suffer from improper handling of the environment terminal states. This introduces implicit rewards priors which can either improve or degrade policy performance.

---

[1]`https://github.com/google-research/google-research/tree/master/dac`
[*]Work done as an intern at Google Brain. Corresponding author: kostrikov@cs.nyu.edu
[†]Work done as a member of the Google AI Residency program (g.co/airesidency)

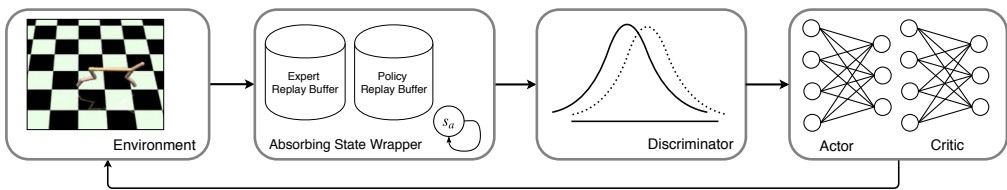

Figure 1: The Discriminator-Actor-Critic imitation learning framework combined with a method to explicitly learn rewards for the absorbing states.

While GAIL requires as little as 200 expert frame transitions (from 4 expert trajectories) to learn a robust reward function on most MuJoCo (Todorov et al., 2012) tasks, the number of policy frame transitions sampled from the environment can be as high as 25 million in order to reach convergence. If PPO (Schulman et al., 2017) is used in place of TRPO (Schulman et al., 2015), the sample complexity can be improved (for example, as in Figure 3, 25 million steps reduces to approximately 10 million steps), however it is still intractable for many robotics or real-world applications. In this work we address this issue by incorporating an off-policy RL algorithm (TD3 (Fujimoto et al., 2018)) and an off-policy discriminator to dramatically decrease the sample complexity by orders of magnitude.

In this work, we also illustrate how specific design choices for AIL algorithms and MDPs used in practice, have a large impact on agent performance for environments with absorbing states. For instance, as we will demonstrate, if the implementation assigns zero rewards for absorbing states, a strictly positive reward function can prevent the agent from solving tasks with a minimal number of steps, while a strictly negative reward function is unable to emulate a survival bonus. Therefore, one must have some knowledge of the true environment reward and incorporate such priors to choose a suitable reward function for successful application of GAIL and AIRL. We will discuss these issues formally, and present a simple - yet effective - solution that drastically improves policy performance for environments with absorbing states; we explicitly handle absorbing state transitions by learning the reward associated with these states.

First we propose a new algorithm, which we call Discriminator-Actor-Critic (DAC) (Figure 1), that is compatible with the GAIL and AIRL frameworks by extending them with an off-policy discriminator and an off-policy actor-critic reinforcement learning algorithm. Then we propose a general approach to handling absorbing states in inverse reinforcement learning and reward learning methods. We experimentally demonstrate that this removes the bias due to incorrect absorbing state handling in both GAIL-like and AIRL-like variants of our DAC algorithm. In our experiments, we demonstrate that DAC achieves state-of-the-art AIL performance for a number of difficult imitation learning tasks, where proper handling of terminal states is crucial for matching expert performance in the presence of absorbing states. More specifically, in this work we:

- Identify, and propose solutions for the problem of handling terminal states of policy rollouts in standard RL benchmarks in the context of AIL algorithms.
- Accelerate learning from demonstrations by providing an off-policy variant for AIL algorithms, which significantly reduces the number of agent-environment interactions.
- Illustrate the robustness of DAC to noisy, multi-modal and constrained expert demonstrations, by performing experiments with human demonstrations on non-trivial robotic tasks.

## 2 RELATED WORK

Imitation learning has been broadly studied under the twin umbrellas of Behavioral Cloning (BC) (Bain & Sommut, 1999; Ross et al., 2011) and Inverse Reinforcement Learning (IRL) (Ng & Russell, 2000). To recover the underlying policy, IRL performs an intermediate step of estimating the reward function followed by RL on this function (Abbeel & Ng, 2004; Ratliff et al., 2006). Operating in the Maximum Entropy IRL formulation (Ziebart et al., 2008), Finn et al. (2016b) introduce an iterative-sampling based estimator for the partition function, deriving an algorithm for recovering non-linear reward functions in high-dimensional state and action spaces. Finn et al. (2016a) and Fu et al.

(2017) further extend this by exploring the theoretical and practical considerations of an adversarial IRL framework, and draw connections between IRL and cost learning in GANs (Goodfellow et al., 2014).

In practical scenarios, we are often interested in recovering the expert's policy, rather than the reward function. Following Syed et al. (2008), and by treating imitation learning as an occupancy matching problem, Ho & Ermon (2016) proposed a Generative Adversarial Imitation Learning (GAIL) framework for learning a policy from demonstrations, which bypasses the need to recover the expert's reward function. More recent work extends the framework by improving on stability and robustness (Wang et al., 2017; Kim & Park, 2018) and making connections to model-based imitation learning (Baram et al., 2017). These approaches generally use on-policy algorithms for policy optimization, trading off sample efficiency for training stability.

Learning complex behaviors from sparse reward signals poses a significant challenge in reinforcement learning. In this context, expert demonstrations or template trajectories have been successfully used (Peters & Schaal, 2008) for initializing RL policies. There has been a growing interest in combining extrinsic sparse reward signals with imitation learning for guided exploration (Zhu et al., 2018; Kang et al., 2018; Le et al., 2018; Vecerík et al., 2017). Off policy learning from demonstration has been previously studied under the umbrella of accelerating reinforcement learning by structured exploration (Nair et al., 2017; Hester et al., 2017) An implicit assumption of these approaches is access to demonstrations and reward from the environment; our approach requires access only to expert demonstrations.

Biases associated with specific MDP benchmarks also arise in the standard RL setup. In particular, Pardo et al. (2017) and Tucker et al. (2018) discuss handling of time limits in RL specifically with MDPs where time limits make the problems non-Markovian and might affect optimality of the training policy and value function estimation. The problem with the biases associated with episode terminations also prove to be severe for AIL algorithms because for specific RL benchmarks the absorbing states might not even be adequately taken into consideration. We discuss this in more detail in Section 4.1.

Our work is most related to AIL algorithms (Ho & Ermon, 2016; Fu et al., 2017; Torabi et al., 2018). In contrast to Ho & Ermon (2016) which assumes *(state-action-state')* transition tuples, Torabi et al. (2018) has weaker assumptions, by relying only on observations and removing the dependency on actions. The contributions in this work are complementary (and compatible) to Torabi et al. (2018).

Concurrent to our work, several other papers introduced algorithms for sample efficient imitation learning. Blondé & Kalousis (2018) introduced Sample-efficient Adversarial Mimic (SAM) algorithm that combines Deep Deterministic Policy Gradients (DDPG) from Lillicrap et al. (2015) with GAIL. While Reddy et al. (2019) and Sasaki et al. (2019) proposed imitation learning algorithms based on off-policy reinforcement learning that does not require to learn rewards.

## 3 BACKGROUND

### 3.1 MARKOV DECISION PROCESS

We consider problems that satisfy the definition of a Markov Decision Process (MDP), formalized by the tuple: $(\mathcal{S}, \mathcal{A}, p(s), p(s'|s, a), r(s, a, s'), \gamma)$. Here $\mathcal{S}$, $\mathcal{A}$ represent the state and action spaces respectively, $p(s)$ is the initial state distribution, $p(s'|s, a)$ defines environment dynamics represented as a conditional state distribution, $r(s, a, s')$ is reward function and $\gamma$ the return discount factor.

In continuing tasks, where environment interactions are unbounded in sequence length, the returns for a trajectory $\tau = \{(s_t, a_t)\}_{t=0}^{\infty}$, are defined as $R_t = \sum_{k=t}^{\infty} \gamma^{k-t} r(s_k, a_k, s_{k+1})$. In order to use the same notation for tasks with absorbing states, whose finite length episodes end when reaching a terminal state, we can define a set of *absorbing states* $s_a$ (Sutton et al., 1998) that an agent enters after the end of episode, has zero reward and transitions to itself for all agent actions: $s_a \sim p(\cdot|s_T, a_T)$, $r(s_a, \cdot, \cdot) = 0$ and $s_a \sim p(\cdot|s_a, \cdot)$ (see Figure 2). With this above absorbing state notation, returns can be defined simply as $R_t = \sum_{k=t}^{T} \gamma^{k-t} r(s_k, a_k, s_{k+1})$. In reinforcement learning, the goal is to learn a policy that maximizes expected returns.

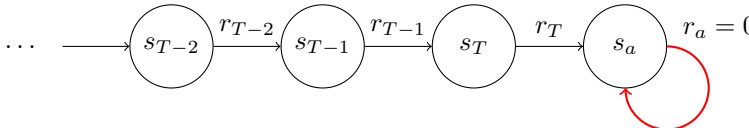

Figure 2: We depict an episode of MDP with an absorbing state. The absorbing state transitions to itself with zero reward.

In many imitation learning and IRL algorithms a common assumption is to assign zero reward value, often implicitly, to absorbing states. Moreover, standard benchmark MDPs, such as the tasks in OpenAI Gym, omit absorbing states and corresponding transitions from rollouts. Under this omission and a de-facto reward of 0 to absorbing states, the standard AIL algorithms do not have access to absorbing states in the buffer, which biases the reward learning process. We propose a modification that enables our DAC algorithm to assign a learned, potentially non-zero, reward for absorbing states. We discuss this in detail in Section 4.1, and demonstrate empirically in Section 5.2 that it is extremely important to properly handle the absorbing states for algorithms where rewards are learned.

Considering the implications of adequate handling of terminal states, it is worth mentioning that practical implementations of MDP benchmarks terminate episodes after a specific number of steps. We refer to this as *time dependent* termination, which makes the tasks non-Markovian, since the returns are now time-dependent as observed in Pardo et al. (2017), Tucker et al. (2018). These works propose to fix this problem by using a time-dependent value function, or by bootstrapping after the terminal state instead of masking the returns, which can be achieved using an algorithm that incorporates value function learning (Fujimoto et al., 2018). Because our solution is derived for infinite horizon problems, we do not treat states that occur after time-dependent termination as absorbing states and assume that after explicitly adding absorbing states and transitions all tasks have infinite horizon (for example, see Figure 2). For this reason, in our implementation we use the latter approach and perform bootstrapping for the terminal states (for elaborate discussion on time limits in MDPs, we refer the reader to Pardo et al. (2017)).

## 3.2 ADVERSARIAL IMITATION LEARNING

In order to learn a robust reward function we use the GAIL framework (Ho & Ermon, 2016). Inspired by maximum entropy IRL (Ziebart et al., 2008) and Generative Adversarial Networks (GANs) (Goodfellow et al., 2014), GAIL trains a binary classifier, $D(s, a)$, referred to as the *discriminator*, to distinguish between transitions sampled from an expert and those generated by the trained policy. In standard GAN frameworks, a generator gradient is calculated by backprop through the learned discriminator. However, in GAIL the policy is instead provided a *reward* for confusing the discriminator, which is then maximized via some on-policy RL optimization scheme (e.g. TRPO (Schulman et al., 2015)):

$$\min_{\pi} \max_{D} \mathbb{E}_{\pi}[\log(D(s, a))] + \mathbb{E}_{\pi_E}[\log(1 - D(s, a))] - \lambda H(\pi) \tag{1}$$

where $H(\pi)$ is an entropy regularization term and $\pi_E$ is a policy provided by an expert.

The rewards learned by GAIL might not correspond to a true reward (Fu et al., 2017) but can be used to match the expert occupancy measure, which is defined as $\rho_{\pi_E}(s, a) = \sum_{t=0}^{\infty} \gamma^t p(s_t = s, a_t = a | \pi_E)$. Ho & Ermon (2016) draw analogies between distribution matching using GANs and occupancy matching with GAIL. They demonstrate that by maximizing the above reward, the algorithm matches occupancy measures of the expert and trained policies with some regularization term defined by the choice of GAN loss function.

In principle, GAIL can be incorporated with any on-policy RL algorithm. However, in this work we adapt it for off-policy training (discussed in Section 4.3). As can be seen from Equation 1, the algorithm requires state-action pairs to be sampled from the learned policy. In Section 4.3 we will discuss what modifications are necessary to adapt the algorithm to off-policy training.

# 4 DISCRIMINATOR-ACTOR-CRITIC

In this section we first elaborate on specific instances of biased rewards in AIL algorithms due to insufficient handling of terminal states. Following that in Section 4.2, we present an approach for unbiasing rewards for existing AIL algorithms. Then, we derive an off-policy formulation of AIL in Section 4.3, which we name Discriminator-Actor-Critic (DAC). A high level pictorial representation of this algorithm is shown in Figure 1, and it is formally summarized in Appendix A.

## 4.1 BIAS IN REWARDS

In the following section, we present examples of bias present in implementations of different AIL algorithms as they assign zero rewards to absorbing states:

- *Absorbing states in MDPs*: In the GAIL framework (and follow-up methods, such as GM-MIL (Kim & Park, 2018), OptionGAN (Henderson et al., 2017a), AIRL and the widely used implementation of GAIL from OpenAI Baselines (Dhariwal et al., 2017)), for some benchmarks such as MuJoCo locomotion tasks from OpenAI Gym, a reward function $r(s, a)$ assigns rewards to intermediate states depending on properties of a task. At the same time, policies executed on these MDPs generate rollouts that ignore absorbing states. Subsequently, the algorithms do not have access to these absorbing states in the buffer, cannot learn proper rewards, and therefore do not perform bootstrapping after terminal states; thus, 0 reward is implicitly assigned for absorbing states.

- For certain environments, a survival bonus in the form of per-step positive reward is added to the rewards received by the agent. This encourages agents to survive longer in the environment to collect more rewards. We observe that a commonly used form of the reward function: $r(s, a) = -\log(1 - D(s, a))$ has worked well for environments that require a survival bonus. Under the implicit assumption of zero rewards for absorbing states in the MDP implementation, this strictly positive estimator cannot recover the true reward function for environments where an agent is required to solve the task as quickly as possible. Using this form of the reward function will lead to sub-optimal solutions. The agent is now incentivized to move in loops or take small actions (in continuous action spaces) that keep it close to the states in the expert's trajectories. The agent keeps collecting positive rewards without actually attempting to solve the task demonstrated by the expert.[2]

- Another reward formulation is $r(s, a) = \log(D(s, a))$. This is often used for tasks with a per step penalty, when a part of a reward function consists of a negative constant assigned unconditionally of states and actions. However, this variant assigns only negative rewards and cannot learn a survival bonus. Such strong priors might lead to good results even with no expert trajectories (as shown in Figure 4).

From an end-user's perspective, it is undesirable to have to craft a different reward function for every new task. In the next section, we propose a method to handle absorbing states of the standard benchmark MDPs in such a way that AIL algorithms are able to recover different reward functions without adjusting the form of reward estimator.

## 4.2 UNBIASING REWARDS

In order to resolve the issues described in Section 4.1, we suggest explicitly learning rewards for absorbing states for expert demonstrations and trajectories produced by a policy. Thus, the returns for final states of an episode that consists of $T$ transitions are defined now $R_T = r(s_T, a_T) + \sum_{t=T+1}^{\infty} \gamma^{t-T} r(s_a, \cdot)$ with a learned reward $r(s_a, \cdot)$ instead of just $R_T = r(s_T, a_T)$ that is often used due to issues described in Section 4.2. This formulation allows the algorithms to correctly estimate returns for the final transitions and optimize the policy accordingly.

In order to enable the AIL algorithms to learn the rewards for absorbing states and RL algorithms to take into account learned rewards, we suggest to update rollouts sampled from MDPs in the following way. After terminating an episode, we explicitly add a transition from the terminal state of the episode to an absorbing state $(s_T, s_a)$ and a transition from an absorbing state to itself $(s_a, s_a)$.

---

[2]Note that this behavior was described in the early reward shaping literature (Ng et al., 1999).

Thus, when sample from the replay buffer AIL algorithms will be able to see absorbing states there were previous hidden, while RL algorithms will be able to properly estimate values for terminal states using transitions $(s_T, s_a)$ and $(s_a, s_a)$ using the following recursions:

$$Q(s_T, a) = r(s_T, a) + \gamma Q(s_a, \cdot)$$
$$Q(s_a, \cdot) = r(s_a, \cdot) + \gamma Q(s_a, \cdot)$$

We implemented these absorbing states by adding an extra indicator dimension that indicates whether the state is absorbing or not, for absorbing states we set the indicator dimension to one and all other dimensions to zero. The GAIL discriminator can distinguish whether reaching an absorbing state is a desirable behavior from the expert's perspective and assign the rewards accordingly.

### 4.3 ADDRESSING SAMPLE INEFFICIENCY

As previously mentioned, GAIL requires a significant number of interactions with a learning environment in order to imitate an expert policy. To address the sample inefficiency of GAIL, we use an off-policy RL algorithm and perform off-policy training of the GAIL discriminator performed in the following way: instead of sampling trajectories from a policy directly, we sample transitions from a replay buffer $\mathcal{R}$ collected while performing off-policy training:

$$\max_D \mathbb{E}_{\mathcal{R}}[\log(D(s, a))] + \mathbb{E}_{\pi_E}[\log(1 - D(s, a))] - \lambda H(\pi). \tag{2}$$

Equation 2 tries to match the occupancy measures between the expert and the distribution induced by the replay buffer $\mathcal{R}$, which can be seen as a mixture of all policy distributions that appeared during training, instead of the latest trained policy $\pi$. In order to recover the original on-policy expectation, one needs to use importance sampling:

$$\max_D \mathbb{E}_{\mathcal{R}}\left[\frac{p_{\pi_\theta}(s, a)}{p_{\mathcal{R}}(s, a)} \log(D(s, a))\right] + \mathbb{E}_{\pi_E}[\log(1 - D(s, a))] - \lambda H(\pi). \tag{3}$$

However, it can be challenging to properly estimate these densities and the discriminator updates might have large variance. We found that the algorithm works well in practice with the importance weight omitted.

We use the GAIL discriminator in order to define rewards for training a policy using TD3; we update per-step rewards every time when we pull transitions from the replay buffer using the latest discriminator. The TD3 algorithm provides a good balance between sample complexity and simplicity of implementation and so is a good candidate for practical applications. Additionally, depending on the distribution of expert demonstrations and properties of the task, off-policy RL algorithms can effectively handle multi-modal action distributions; for example, this can be achieved for the Soft Actor Critic algorithm (Haarnoja et al., 2018b) using the reparametrization trick (Kingma & Ba, 2014) with a normalizing flow (Rezende & Mohamed, 2015) as described in Haarnoja et al. (2018a).

## 5 EXPERIMENTS

We implemented the DAC algorithm described in Section 4.3 using TensorFlow Eager (Abadi et al., 2015) and we evaluated it on popular benchmarks for continuous control simulated in MuJoCo (Todorov et al., 2012). We also define a new set of robotic continuous control tasks (described in detail below) simulated in PyBullet (Coumans & Bai, 2016), and a Virtual Reality (VR) system for capturing human examples in this environment; human examples constitute a particularly challenging demonstration source due to their noisy, multi-modal and potentially sub-optimal nature, and we define multi-task environments as a challenging setup for adversarial imitation learning.

For the critic and policy networks we used the same architecture as in Fujimoto et al. (2018): a 2 layer MLP with ReLU activations and 400 and 300 hidden units correspondingly. We also add gradient clipping (Pascanu et al., 2013) to the actor network with clipping value of 40. For the

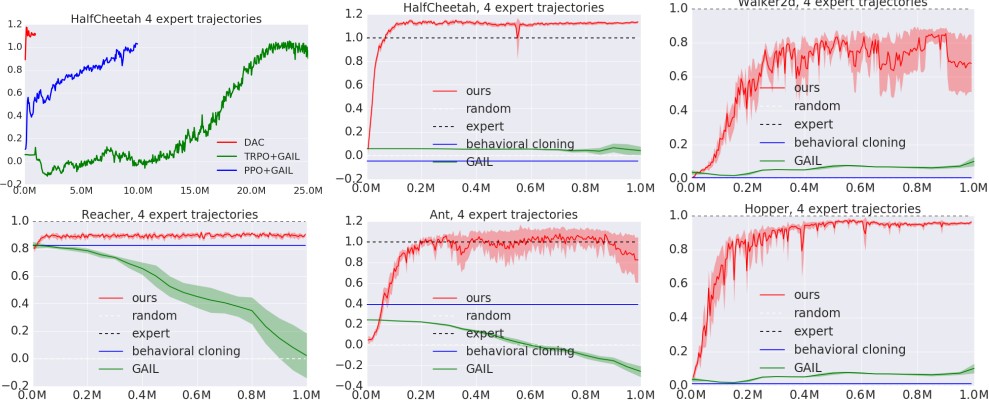

Figure 3: Comparisons of algorithms using 4 expert demonstrations. y-axis corresponds to normalized reward (0 corresponds to a random policy, while 1 corresponds to an expert policy).

discriminator we used the same architecture as in Ho & Ermon (2016): a 2 layer MLP with 100 hidden units and $\tanh$ activations. We trained all networks with the Adam optimizer (Kingma & Ba, 2014) and decay learning rate by starting with initial learning rate of $10^{-3}$ and decaying it by 0.5 every $10^5$ training steps for the actor network.

In order to make the algorithm more stable, especially in the off-policy regime when the discriminator can easily over-fit to training data, we use regularization in the form of gradient penalties (Gulrajani et al., 2017) for the discriminator. Originally, this was introduced as an alternative to weight clipping for Wasserstein GANs (Arjovsky et al., 2017), but later it was shown that it helps to make JS-based GANs more stable as well (Lucic et al., 2017).

We replicate the experimental setup of Ho & Ermon (2016): expert trajectories are sub-sampled by retaining every 20 time steps starting with a random offset (and fixed stride). It is worth mentioning that, as in Ho & Ermon (2016), this procedure is done in order to make the imitation learning task harder. With full trajectories, behavioral cloning provides competitive results to GAIL.

Following Henderson et al. (2017b) and Fujimoto et al. (2018), we perform evaluation using 10 different random seeds. For each seed, we compute average episode reward using 10 episodes and running the policy without random noise. As in Ho & Ermon (2016) we plot reward normalized in such a way that zero corresponds to a random reward while one corresponds to expert rewards. We compute mean over all seeds and visualize half standard deviations. In order to produce the same evaluation for GAIL we used the original implementation[3] of the algorithm.

## 5.1 OFF POLICY DAC ALGORITHM

Evaluation results of the DAC algorithm on a suite of MuJoCo tasks are shown in Figure 3, as are the GAIL (TRPO) and BC basline results. In the top-left plot, we show DAC is an order of magnitude more sample efficent than then TRPO and PPO based GAIL baselines. In the other plots, we show that by using a significantly smaller number of environment steps (orders of magnitude fewer), our DAC algorithm reaches comparable expected reward as the GAIL baseline. Furthermore, DAC outperforms the GAIL baseline on all environments within a 1 million step threshold. We obtained slightly worse results for Walker2d. However, as mentioned earlier, GAIL uses a reward function that already has some biases encoded in it that aids training on this specific environment. A comprehensive suit of results can be found in Appendix B, Figure 7.

## 5.2 REWARD BIAS

As discussed in Section 4.1, the reward function variants used with GAIL can have implicit biases when used without handling absorbing states. Figure 4 demonstrates how bias affects results on an

---

[3]https://github.com/openai/imitation

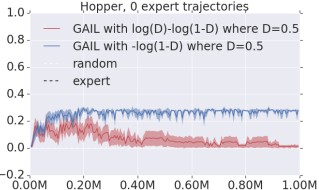

Figure 4: Even without training, some reward functions can perform well on some tasks.

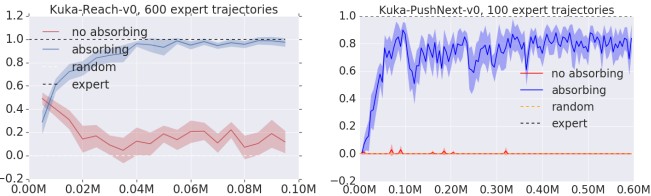

Figure 5: Effect of absorbing state handling on Kuka environments. For these environments, we use human demonstrations as expert trajectories, and GAIL framework with a positive reward function.

environment with survival bonus when using the reward function of Ho & Ermon (2016): $r(s, a) = -\log(1 - D(s, a))$. Surprisingly, when using a fixed and untrained GAIL discriminator that outputs 0.5 for every state-action pair, we were able to reach episode rewards of around 1000 on the Hopper environment, corresponding to approximately one third of the expert performance. Without any reward learning, and using no expert demonstrations, the agent can learn a policy that outperforms behavioral cloning (Figure 4). Therefore, the choice of a specific reward function might already provide strong prior knowledge that helps the RL algorithm to move towards recovering the expert policy, irrespective of the quality of the learned reward.

Additionally, we evaluated our method on two environments with per-step penalty (see Figure 5). These environment are simulated in PyBullet and consist of a Kuka IIWA arm and 3 blocks on a virtual table. A rendering of the environment can be found in Appendix C, Figure 8. Using a Cartesian displacement action for the gripper end-effector and a compact observation-space (consisting of each block's 6DOF pose and the Kuka's end-effector pose), the agent must either a) reach one of the 3 blocks in the shortest number of frames possible (the target block is provided to the policy as a one-hot vector), which we call *Kuka-Reach*, or b) push one block along the table so that it is adjacent to another block, which we call *Kuka-PushNext*. For evaluation, we define a sparse reward indicating successful task completion (within some threshold). For these imitation learning experiments, we use human demonstrations collected with a VR setup, where the participant wears a VR headset and controls in real-time the gripper end-effector using a 6DOF controller.

Using the reward defined as $r(s, a) = -log(1 - D(s, a))$ and without absorbing state handling, the agent completely fails to recover the expert policy given 600 expert trajectories without sub-sampling (as shown in Figure 4). In contrast, our DAC algorithm quickly learns to imitate the expert, despite using noisy and potentially sub-optimal human demonstrations.

As discussed, alternative reward functions do not have this positive bias but still require proper handling of the absorbing states as well in order to avoid early termination due to incorrectly assigned per-frame penalty. Figure 6 illustrates results for AIRL with and without learning rewards for absorbing states. For these experiments we use the discriminator structure from Fu et al. (2017) in combination with the TD3 algorithm.

## 6 CONCLUSION

In this work we address several important issues associated with the popular GAIL framework. In particular, we address 1) sample inefficiency with respect to policy transitions in the environment and 2) we demonstrate a number of reward biases that can either implicitly impose prior knowledge about the true reward, or alternatively, prevent the policy from imitating the optimal expert. To

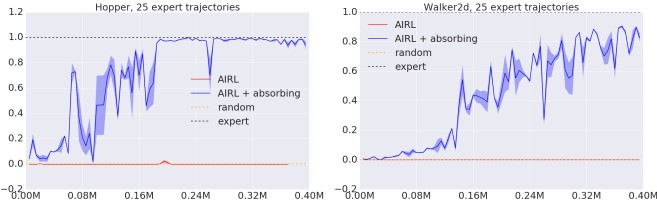

Figure 6: Effect of learning absorbing state rewards when using an AIRL discriminator within the DAC Framework in OpenAI Gym environments.

address reward bias, we propose a simple mechanism whereby the rewards for absorbing states are also learned, which negates the need to hand-craft a discriminator reward function for the properties of the task at hand. In order to improve sample efficiency, we perform off-policy training of the discriminator and use an off-policy RL algorithm. We show that our algorithm reaches state-of-the-art performance for an imitation learning algorithm on several standard RL benchmarks, and is able to recover the expert policy given a significantly smaller number of samples than in recent GAIL work.

ACKNOWLEDGMENTS

We would like to thank Justin Fu and the ICLR Reproducibility Workshop team for insightful discussions and feedback.

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

# A  DAC ALGORITHM

---

**Algorithm 1** Discriminative-Actor-Critic Adversarial Imitation Learning Algorithm

---

**Input**: expert replay buffer $\mathcal{R}_E$

  **procedure** WRAPFORABSORBINGSTATES($\tau$)

    **if** $s_T$ is a terminal state not caused by time limits **then**

      $\tau \leftarrow \tau \setminus \{(s_T, a_T, \cdot, s'_T)\} \cup \{(s_T, a_T, \cdot, s_a)\}$

      $\tau \leftarrow \tau \cup \{(s_a, \cdot, \cdot, s_a)\}$

    **end if**

    **return** $\tau$

  **end procedure**

 

  Initialize replay buffer $\mathcal{R} \leftarrow \emptyset$

  **for** $\tau = \{(s_t, a_t, \cdot, s'_t)\}_{t=1}^T \in \mathcal{R}_E$ **do**

    $\tau \leftarrow$ WrapForAbsorbingState($\tau$)                  ▷ Wrap expert rollouts with absorbing states

  **end for**

  **for** $n = 1, \ldots,$ **do**

    Sample $\tau = \{(s_t, a_t, \cdot, s'_t)\}_{t=1}^T$ with $\pi_\theta$

    $\mathcal{R} \leftarrow \mathcal{R} \cup$ WrapForAbsorbingState($\tau$)          ▷ Update Policy Replay Buffer

    **for** $i = 1, \ldots, |\tau|$ **do**

      $\{(s_t, a_t, \cdot, \cdot)\}_{t=1}^B \sim \mathcal{R},\;\; \{(s'_t, a'_t, \cdot, \cdot)\}_{t=1}^B \sim \mathcal{R}_E$      ▷ Mini-batch sampling

      $\mathcal{L} = \sum_{b=1}^B -\log D(s_b, a_b) - \log(1 - D(s'_b, a'_b))$

      Update $D$ with GAN+GP

    **end for**

    **for** $i = 1, \ldots, |\tau|$ **do**

      $\{(s_t, a_t, \cdot, s'_t)\}_{t=1}^B \sim \mathcal{R}$

      **for** $b = 1, \ldots, B$ **do**

        $r \leftarrow \log D(s_b, a_b) - \log(1 - D(s_b, a_b))$

        $(s_b, a_b, \cdot, s'_b) \leftarrow (s_b, a_b, r, s'_b)$          ▷ Use current reward estimate.

      **end for**

      Update $\pi_\theta$ with TD3

    **end for**

  **end for**

---

# B    SUPPLEMENTARY RESULTS ON MUJOCO ENVIRONMENTS

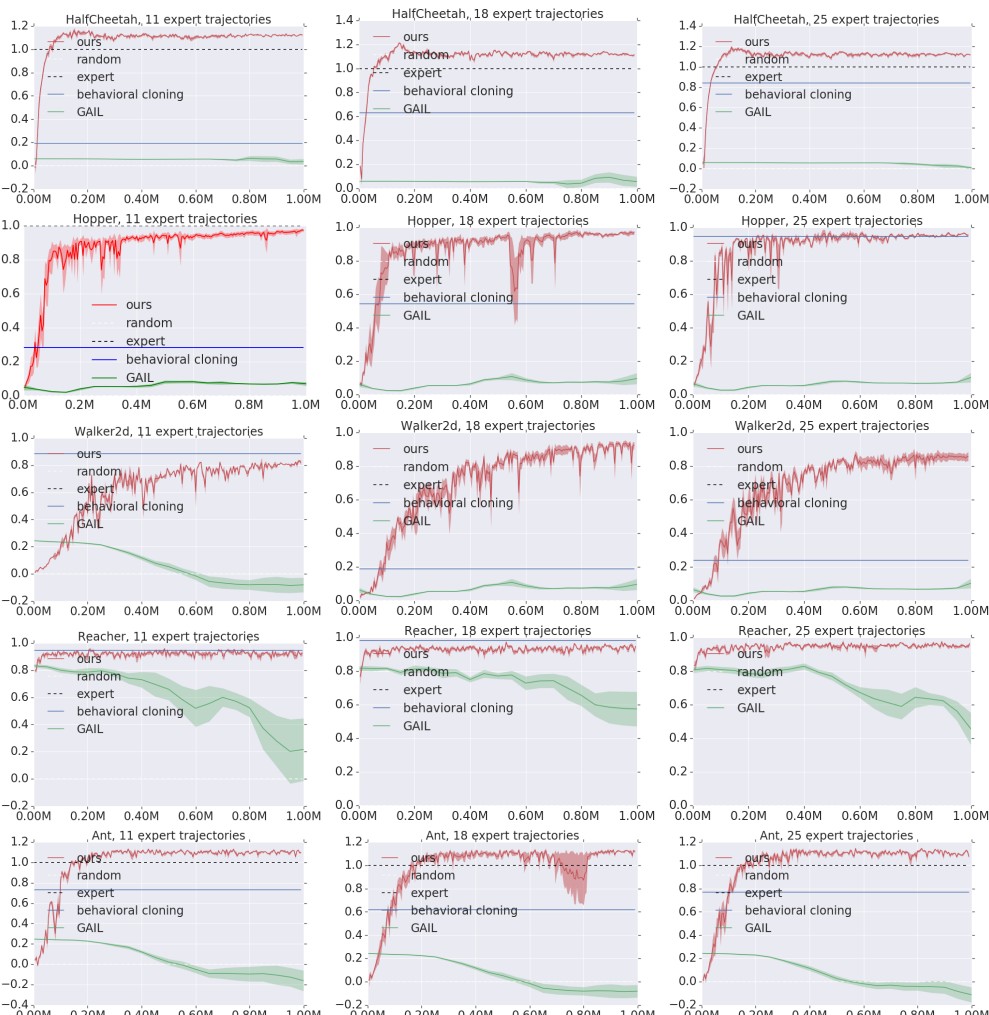

Figure 7: Comparisons of different algorithms given the same number of expert demonstrations. y-axis corresponds to normalized reward (0 corresponds to a random policy, while 1 corresponds to an expert policy).

## C KUKA-IIWA SIMULATED ENVIRONMENT

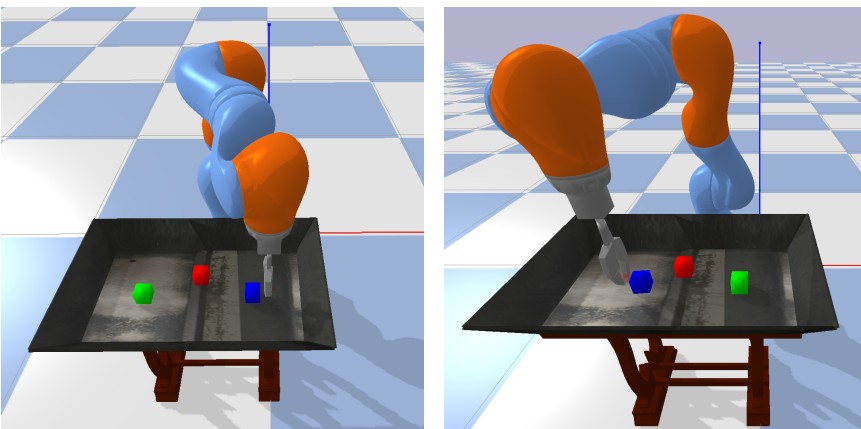

Figure 8: Renderings of our Kuka-IIWA environment. Using a VR headset and 6DOF controller, a human participant can control the 6DOF end-effector pose in order to record expert demonstrations. In the *Kuka-Reach* tasks, the agent must bring the robot gripper to 1 of the 3 blocks (where the state contains a 1-hot encoding of the task) and for the *Kuka-PushNext* tasks, the agent must use the robot gripper to push one block next to another.

