# OpenReview forum: "Discriminator-Actor-Critic: Addressing Sample Inefficiency and Reward Bias in Adversarial Imitation Learning"
_ICLR.cc/2019/Conference_

### Official Review · AnonReviewer3 · 2018-11-01
**Interesting paper on the challenges of GAIL**

**Rating:** 7
**Confidence:** 3

**Review:**

This paper investigates two issues regarding Adversarial Imitation Learning. They identify a bias in commonly used reward functions and provide a solution to this. Furthermore they suggest to improve sample efficiency by introducing a off-policy algorithm dubbed "Discriminator-Actor-Critic". They key point here being that they propose a replay buffer to sample transitions from.

It is well written and easy to follow. The authors are able to position their work well into the existing literature and pointing the differences out.

Pros:
	* Well written
	* Motivation is clear
	* Example on biased reward functions
	* Experiments are carefully designed and thorough
Cons:
	* The analysis of the results in section 5.1 is a bit short

Questions:
	* You provide a pseudo code of you method in the appendix where you give the loss function. I assume this corresponds to Eq. 2. Did you omit the entropy penalty or did you not use that termin during learning?

	* What's the point of plotting the reward of a random policy? It seems your using it as a lower bound making it zero. I think it would benefit the plots if you just mention it instead of plotting the line and having an extra legend

	* In Fig. 4 you show results for DAC, TRPO, and PPO for the HalfCheetah environment in 25M steps. Could you also provide this for the remaining environments?

	* Is it possible to show results of the effect of absorbing states on the Mujoco environments?

Minor suggestions:
In Eq. (1) it is not clear what is meant by pi_E. From context we can assume that E stands for expert policy. Maybe add that. Figures 1 and 2 are not referenced in the text and their respective caption is very short. Please reference them accordingly and maybe add a bit of information. In section 4.1.1 you reference figure 4.1 but i think your talking about figure 3.

---

> ### Author Response · Authors · 2018-11-14
> **Response to AnonReviewer3**
>
> We thank the reviewer for the positive and constructive feedback.
>
> We have extended the section 5.1 of the manuscript as suggested by the reviewer.
>
> Below are detailed answers for the reviewer’s concerns:
>
> 1) To simplify the exposition we omitted the entropy penalty as it does not contribute meaningfully to the algorithm performance in our experimentation. Similar findings were observed in the GAIL paper, where the authors disregarded the entropy coefficient for every tested environment, except for the Reacher environment.
>
> 2) We added the performance of a random policy to the graph to be consistent with the original GAIL paper. We believe that it improves readability of the plot by providing necessary scaling.
>
> 3) We already started working on additional experimentation as requested. We will update the manuscript as soon as we gather these results.
>
> 4) We observed the same effect of having absorbing states in the Kuka arm tasks (Fig. 6), as in the MuJoCo environments. Also, we evaluated absorbing states within the AIRL framework for Walker-2D and Hopper environments (Fig. 7). We demonstrate that proper handling of absorbing states is critical for effectively imitating the expert policy.
>
> In addition, we updated the paper to accommodate the minor suggestions proposed by the reviewer.

---

### Official Review · AnonReviewer2 · 2018-11-03
**Sound and effective approach with little novelty, insufficient analysis of reward bias**

**Rating:** 6
**Confidence:** 4

**Review:**

The paper suggests to use TD3 to compute an off-policy update instead of the TRPO/PPO updates in GAIL/AIRL in order to increase sample efficiency.
The paper further discusses the problem of implicit step penalties and survival bias caused by absorbing states, when using the upper-bounded/lower-bounded reward functions log(D) and -(1-log(D)) respectively. To tackle these problem, the paper proposes to explicit add a unique absorbing state at the end of each trajectory, such that its rewards can be learned as well.

Pro:
The paper is well written and clearly presented.

Using a more sample efficient RL method for the policy update is sensible and turned out effective in the experiments.

Properly handling simulator resets in MDPs is a well known problem in reinforcement learning that I think is insufficiently discussed in the context of IRL.


Cons:
The contributions seem rather small.
a) Replacing the policy update is trivial, since the rl methods are used as black-box modules for the discussed AIL methods.

b) Using importance weighting to reuse old trajectories for the discriminator update hardly counts as a contribution either--especially when the importance weights are simply omitted in practice. I also think that the reported problems due to the high variance have not been sufficiently investigated. There should be a better solution than just pretending that the replay buffer corresponds to roll-outs of the current policy. Would it maybe help to use self-normalized importance weights? The paper does also not analyze how such assumption/approximation affects the theoretical guarantees.

c) The problem with absorbing states is in my opinion the most interesting contribution of the paper. However, the discussion is rather shallow and I do not think that the illustrative example is very convincing. Section 4.1.1. argues that for the given policy roll-out, the discriminator reward puts more reward on the policy trajectory than the expert trajectory. However, it is neither surprising nor problematic that the discriminator reward does not produce the desired behavior during learning. By assigning more cumulative reward for s2_a1->s1 than for s2_a2->g, the policy would (after a few more updates) choose the latter action much less frequently than with probability 0.5 and the corresponding reward would grow towards infinity until at some point Q(s2,a2) > Q(s2,a1)--when the policy would match the expert exactly. The illustrative example also uses more policy-labeled transitions than agent-labeled ones for learning the classifier, which may also be problematic. The paper further argues that a strictly positive reward function always rewards a policy for avoiding absorbing states, which I think is not true in general. A strictly positive reward function can still produce arbitrary large reward for any action that reaches an absorbing state. Hence, the immediate reward for choosing such action can be made larger than the discounted future reward when not ending the episode (for any gamma < 1). Even for state-only reward functions the problem does not persist when reseting the environment after reaching the absorbing state such that the training trajectories contain states that are only reached if the simulator gets reset. Hence, I am not convinced that adding a special absorbing state to the trajectory is necessary if the simulation reset is correctly implemented. This may be different for resets due to time limits that can not be predicted by the last state-action tuple. However, issues relating to time limits are not addressed in the paper. I also think that it is strange that the direct way of computing the return for the terminal state is much less stable than recursively computing it and think that the paper should include a convincing explanation.

---------------
Update 21.11.2018

I think my initial assessment was too positive. During the rebuttal, I noticed that the discussion of reward bias was not only shallow but also wrong in some aspects and very misleading, because problems arising from hacky implementations of some RL toolboxes were discussed as theoretical shortcoming of AIL algorithms. Hence, I think the initial submission should be clearly rejected. However, the authors submitted a revised version that presents the root of the observed problem much more accurately. I think that the revised version is substantially better than the original submission. However, I think that my initial rating is still valid (better: became valid), because the main issues that I raised for the initial submission still apply to the current revision, namely:
- The technical contributions are minor.
- The theoretical discussion (in particular regarding absorbing states) is quite shallow.

The merits of the paper are:
- Good results due to off-policy learning
- Raising awareness and providing a fix for a common pitfall

I think that the problems arising from incorrectly treated absorbing states needs to be discussed more profoundly.
Some suggestions:

Section 3.1
"As we discuss in detail in Section 4.2 [...]"
I think this should refer to section 4.1. Also the discussion should in section 4.1 should be a bit more detailed. How do common implementations implicitly assign zero rewards? Which implementations are affected? Which papers published inferior results due to this bug? I think it is also important to note, that absorbing states are hidden from the algorithm and that the reward function is thus only applied to non-absorbing states.

"We will demonstrate empirically in Section 4.1 [...]"
The demonstration is currently missing. I think it would be nice to illustrate the problem on a simple example. The original example might actually work, as shown by the code example of the rebuttal, however the explanation was not convincing. Maybe it would be easier to argue with a simpler algorithm (e.g MaxEnt-IRL, potentially projecting the rewards to positive values)?

Section 3.1 seems to focus too much on resets that are caused by time limits. Such resets are inherently different from terminal states such as falling down in locomotion tasks, because they can not be modelled with the given MDP formulation unless time is considered part of the state. Indeed, I think that for infinite horizon MDPs without time-awareness, time limits can not be modelled using absorbing states (I think the RL book misses to mention that time needs to be part of the state such that the policy remains Markovian, which is a bit misleading). Instead those resets are often handled by returning an estimate of the future return (bootstrapping). This treatment of time limits is already part of the TD3 implementation and as far as I understood not the focus of the paper. Instead section 3.1. should focus on resets caused by task failure/completion, which can actually be modelled with absorbing states, because the agent will always transition to the absorbing state when a terminal state is reached which is in line with Markovian dynamics.

Section 4.2 should also add a few more details. Did I understand correctly, that when computing the return R_T the sum is indeed finite and stopped after a fixed horizon? If yes, this should be reflected in the equation, and the horizon should be mentioned in the paper. The paper should also better explain how  the proposed fix enables the algorithm to learn the reward of the absorbing state. For example, section 4.2. does not even mention that the state s_a was added as part of the solution.


-------------
Update 22.11.2018
By highlighting the difference between termination due to time-limits and termination due to task completion, and by better describing how the proposed fix addresses the problem of reward bias that is present in common AIL implementations, the newest revision further improves the submission.
I think that the submission can get accepted and I adapted my rating accordingly.

Minor:
Conclusion should also squeeze in somehow that the reward biases are caused by the implementations.
Typo in 4.2: "Thus, when sample[sic] from the replay buffer AIL algorithms will be able to see absorbing states there[sic]
were previous hidden, [...]"

---

> ### Author Response · Authors · 2018-11-14
> **Response to AnonReviewer2**
>
> We thank the reviewer for the detailed and constructive feedback. We address the above mentioned points and add some additional experiments, as detailed below.
>
> c) “By assigning more cumulative reward for s2_a1->s1 than for s2_a2->g, the policy would (after a few more updates) choose the latter action much less frequently than with probability 0.5 and the corresponding reward would grow towards infinity until at some point Q(s2,a2) > Q(s2,a1)--when the policy would match the expert exactly.”
> “The paper further argues that a strictly positive reward function always rewards a policy for avoiding absorbing states, which I think is not true in general. A strictly positive reward function can still produce arbitrary large reward for any action that reaches an absorbing state.”
>
> > This is a good point, and we will discuss this situation in more detail in the final paper. However, we do not believe that this directly applies to adversarial learning algorithms, such as the ones studied in our paper. We provide discussion as well as a numerical example below, which will be included in the paper.
>
> The aforementioned situation can only happen in the limit, but the next discriminator update will return the policy to the previous state, in which it is more advantageous to take a loop, according to the GAIL reward definition. Therefore, the original formulation of the algorithm does not converge in this case. In contrast, learning rewards for the absorbing states will resolve this issue.
>
> Moreover, the example provided by the reviewer assumes that we can fix the reward function at some point of training and then train the policy to optimality according to this reward function; while devising a scheme to early terminate learning of the reward function is possible, it is not specified by the dynamic reward learning mechanisms of the GAIL algorithm, which alternates updates between the policy and the discriminator. Please see a simple script that illustrates the example (anonymous link):
> https://colab.research.google.com/drive/1gV56NLik367nslwK7iJzs8WTe5tD-BO5
>
> This specific toy example will be included into our open source release.
>
> “Hence, I am not convinced that adding a special absorbing state to the trajectory is necessary if the simulation reset is correctly implemented.”
>
> >  Could you please clarify what do you mean by a correct implementation of simulation resets?
>
> “I also think that it is strange that the direct way of computing the return for the terminal state is much less stable than recursively computing it and think that the paper should include a convincing explanation.”
>
> > We think that it is less stable to analytically compute the returns for absorbing states as it introduces a high variance for TD updates of the value network due to the fact that we bootstrapped for all states. The issue is well known and usually solved by using target networks (see https://www.nature.com/articles/nature14236).
>
> “This may be different for resets due to time limits that can not be predicted by the last state-action tuple. However, issues relating to time limits are not addressed in the paper”
>
> > Although some of the benchmark tasks do have an episodic time limit, an off-policy RL algorithm can still calculate a (discounted) target value at the last time step in such environments, which is what our implementation of TD3 actually does. Please see the original implementation of TD3 for more details:
> https://github.com/sfujim/TD3/blob/master/main.py#L123
>
> a) We note that this does make a substantial difference in terms of sample efficiency over prior work on adversarial IL, as shown in Figure 4 -- we believe that such substantial improvements in efficiency are of interest to the ICLR community, though it is not the sole contribution of our paper.
>
> b) We did use normalized importance weights, but unfortunately did not find that the resulting method performed well, while simply omitted importance weights achieved good performance. We think that the naive way of estimating importance weights increases variance of updates. We will analyze this further in the final version, but for now we would emphasize that this is not the primary contribution of the work, but only a technical detail that we discussed for completeness.

---

> > ### Comment · AnonReviewer2 · 2018-11-15
> > **Why I think that the random resets are incorrectly implemented**
> >
> > Let me elaborate on why I think that the failure of the existing methods to match the expert is caused solely by an incorrect implementation of the MDP and are not shortcomings of the actual algorithms.
> >
> > Roll-outs in an MDPs either have a fixed length (finite horizon) or an infinite length (infinite horizon). Variable length trajectory can be simulated by introducing absorbing states in a finite horizon formulation as mentions in section 3.1. of the submission. The infinite horizon case can be approximated using a large horizon and a time-dependent reward function for the discounting. However, in either case the absorbing states need to be treated in the same way as any other state in the MDP. Importantly, these states do not end the episode prematurely but just prevent the agent from entering any non-absorbing state and yield the same value for each policy. In reinforcement learning, we can typically stop the episode and return the Q-Value (which happens to equal the immediate reward, if the constant rewards of absorbing is assumed to be zero) which allows for more efficient implementations. However, it is important to note that the reward function is then only evaluated on the non-absorbing states and the rewards for absorbing states are implicitly assumed to be zero. Hence, when implementing policy roll-outs with a "break" one needs to be aware that the specified reward function does not correspond to the actual reward function of the MDP but affects only a subset of the possible state-action pairs (as those states of the MDP that we call "absorbing" will not be affected). This is well known in reinforcement learning, and even exploited by specifying constant offsets in the reward function for survival bonus / time penalty which would be useless if the specified reward function would be the actual reward function of the MDP.
> >
> > Using such implementation of an environment which is targeted at reinforcement learning and using it for inverse reinforcement learning
> > is incorrect, because IRL algorithms are typically derived for learning the reward function for the whole MDP and not for a subset of the MDP. How can we expect an algorithm to learn the correct constant offset of a reward function (which does affect the optimal policy in the given implementation) using a formulation that implies that an offset does not affect the optimal behaviour?
> >
> > To summarize: The failure of GAIL and AIRL of matching expert demonstrations for some RL toolkits with absorbing states is caused by implementation hacks that are fine for RL problems and specific reward functions, but not for IRL. Indeed, the convergence problem of the code example can be solved simply by implementing the MDP in the way it is defined in section 3.1.--using a (discounted) fixed horizon and absorbing states. My code can be found at https://colab.research.google.com/drive/11w0McKxg7AA6ueTQNbfTtYAyVrSKgU2z
> > The only changes were
> > - adding the missing state (s_g) and transition (sg->sg) to the MDP
> > - removing the break in the roll-out
> > - using a full expert trajectory as demonstration (including absorbing transitions)
> > - solving some numerical issues when both expert and agent have probability 0 for choosing a given action.
> > The algorithm converges reliably to a policy that produces average trajectory lengths of 2.0
> >
> > Of course, the solution of the paper is a bit more elegant since it avoids to simulate the whole trajectory, but the effect should be the same. It is important to raise awareness of such pitfalls, but I do not think that it is enough to write an ICLR paper about--especially if it is discussed as an algorithmic improvement, when the algorithms are just fine.
> >
> > Also in conjunction with the other minor contributions (using all trajectories for training the discriminator--without any theoretical justification, and using a more sample efficient policy update), I don't think that the contributions of the submission are sufficient.

---

> > > ### Author Response · Authors · 2018-11-16
> > > **Additional response to AnonReviewer2**
> > >
> > > Thank you for your detailed response. We generally agree with the technical side of your description: MDPs with absorbing states require the absorbing states to be handled properly for IRL. This is in essence the point of this portion of our paper. We also agree that addressing this is not so much a new algorithm as it is a fix to the MDP. We have edited the paper to reflect this and clarify this point, please see the difference between the last revision and original submission (the abstract, sections 3.1 and 4). The fact that we test our solution by extending two different prior methods (GAIL and AIRL) reflects the generality of the solution.
> > >
> > > However, we respectfully disagree that this solution is obvious or trivial. Environments with absorbing states in the MuJoCo locomotion benchmark tasks have been used as benchmarks for imitation learning and IRL in one form or another for over two years. In this time, no one has corrected this issue, or even noted that this issue exists, and numerous works incorrectly treat absorbing states, resulting in results that are not an accurate reflection of the performance of these algorithms, as detailed in Section 5.2 and Figures 5,6 and 7 of our paper. This issue is severe, it is making it difficult to evaluate IRL and imitation algorithms, and as far as we can tell, most of the community is unaware of it. We believe that our paper will raise awareness of this issue and facilitate the development and evaluation of better IRL algorithms in the future. With your help, we have clarified this point further in our current paper. The purpose of a research paper is to communicate an idea that is relevant and important to a large subset of the community, and we believe that our paper does this.

---

> > > > ### Comment · AnonReviewer2 · 2018-11-19
> > > > **Revised version is still very misleading 1/2**
> > > >
> > > > I agree that communicating an idea that is relevant and important for a large subset of the community can justify publishing a research paper--even if the technical contribution is marginal. However the submission communicates an idea that in my opinion is just wrong. Namely, the submission communicates the idea that existing methods for IRL can not handle absorbing states and that learning reward functions that are always positive/negative can lead to an implicit bias. This is plain wrong and not helping the research community at all. Communicating this idea is not important but can be very harmful, especially when it is published at a conference like ICLR. I don't want to review papers next year that propose fixed offsets in order to enable their reward functions to produce both signs (and the like). I know that we need to sell our stuff and I'm fine with calling a TD3 replacement an all new algorithm. But, discussing a fix for a hack in a toolbox as an algorithmic enhancement just can not work out. The initial submission did not even give a hint that the bias is only caused by hacky implementations of the MDP, but pretended that it results from shortcomings of the algorithms. I agree that the the revised version is much better by admitting that it only applies to specific implementations. However, in order to clearly communicate the actual idea it is not sufficient to add one small paragraph, because the original, harmful narrative pervades the whole paper. I propose a number of modification (split over two comments due to character limits) to the paper, that I think are necessary to communicate to the reader how the improved performance was reached. The contribution of the revised version could be just enough to push it over the acceptance threshold.
> > > >
> > > > Introduction:
> > > > "[...] 2) bias in the reward function formulation and improper handling of environment terminal states introduces implicit rewards priors that can either improve
> > > > or degrade policy performance."
> > > > - This still makes the impression that both AIL methods are biased and can not handle absorbing states correctly.
> > > >
> > > > "In this work we will also illustrate how the specific form of AIL reward function used has a large
> > > > impact on agent performance for episodic environments.  For instance, as we will show, a strictly
> > > > positive reward function prevents the agent from solving tasks in a minimal number of steps and a
> > > > strictly negative reward function is not able to emulate a survival bonus.  Therefore, one must have
> > > > some knowledge of the true environment reward and incorporate such priors to choose a suitable
> > > > reward function for successful application of GAIL and AIRL. We will discuss these issues in formal
> > > > detail, and present a simple - yet effective - solution that drastically improves policy performance
> > > > for episodic environments; we explicitly handle absorbing state transitions by learning the reward
> > > > associated with these states"
> > > > - This paragraph needs to be completely rewritten. The form of the reward function (whether it is strictly positive or negative) does in theory not matter at all. It is completely fine to learn a reward function that only produces positive/negative values. Don't make the impression, that IRL researchers should start looking for ways to learn reward functions that can produce both signs. Furthermore, from a theoretical perspective GAIL and AIRL already explicitly learn rewards associated with absorbing states. This paragraph should clearly state that commonly used implementations of the roll-outs are not in line with the MDP-formulation which may be fine for RL but can lead to problems with IRL approaches. You may already want to point to the "break"-statement and state that it prevents the learned reward function from being applied to absorbing states. Although it is interesting to show how strictly positive/negative reward functions are affected by such implementations and it is nice to discuss these effects in the paper (maybe not in the introduction) and confirm them in the experiment, don't discuss the sign of the reward as the central problem. Also make sure to state, that you propose a different way of implementing the MDPs that allows early termination while fixing this problem. It is in my opinion crucial to discuss the problem and the solution in the context of implementing policy roll-outs / absorbing states. Make sure to show that this is a relevant problem that affects multiple toolboxes and that algorithms were incorrectly evaluated due to this issue - put in some references to undermine your claim that numerous work treat absorbing states incorrectly.

---

> > > > > ### Comment · AnonReviewer2 · 2018-11-19
> > > > > **Revised version is stell very misleading 2/2**
> > > > >
> > > > > "We propose a new algorithm, which we call Discriminator-Actor-Critic (DAC) (see Figure 1), that is
> > > > > compatible with both the popular GAIL and AIRL frameworks, incorporates explicit terminal state
> > > > > handling, an off-policy discriminator and an off-policy actor-critic reinforcement learning algorithm"
> > > > > - Don't say that DAC incorporates terminal state handling. Rather write something like
> > > > >
> > > > > "We propose a new algorithm, which we call Discriminator-Actor-Critic (DAC) (see Figure 1), that extends GAIL and AIRL by replacing the policy update by the more sample efficient TD3 algorithm. Furthermore, our implementation of DAC includes a proper handling of terminal states that can be straightforwardly transferred to other inverse reinforcement learning algorithms. We show in ablative experiments, that our off-policy inverse reinforcement learning approach requires approximately an order of magnitude fewer policy roll-outs than the state of the art, and that proper handling of terminal states is crucial for matching expert demonstrations in the presence of absorbing states."
> > > > >
> > > > > End of Introduction:
> > > > > "• Identify, and propose solutions for the problem of bias in discriminator-based reward estimation in imitation learning."
> > > > > - As far as I can tell, there is no bias is discriminator-based reward estimation. I don't think that the proposed solution has to do with discriminators at all, but would affect any IRL algorithm and all those IL algorithm that use RL in the loop. Change this point to something like "Identify early termination of policy roll-outs in commonly used reinforcement learning toolboxes as cause of reward bias in the context of inverse reinforcement learning and propose a solution that allows to correctly match expert demonstrations in the presence of absorbing states."
> > > > >
> > > > > Related work should discuss prior work related to incorrectly handling absorbing states (in RL or IRL). However, I don't think that there is much published literature about fixing implementation hacks.
> > > > > I'm aware of a paper at last ICML [1] (that was previously rejected for ICLR due to lack of novelty), that discussed problems relating to time-limits in infinite horizon formulations which might be worth mentioning.
> > > > >
> > > > > Section 3 needs to explain how exactly the baselines implementation breaks IRL algorithms for absorbing states. The last paragraph of section 3.1. is not at all sufficient to communicate the root of the problem to the reader. Explain why the break in the roll-out violates the MDP formulation (which is assumed by the discussed algorithm) and that the learned reward function is thus not applied to the MDP. Add a new section (after 3.1 or 3.2) that is at least as detailed as my last comment.
> > > > >
> > > > > Section 4.1. also needs to be rewritten completely. There is no bias for the different reward formulations. Rather, applying IRL/IL algorithms without sufficient care to rl toolboxes that use hacky implementations can lead to different problems for different reward formulations.
> > > > >
> > > > > Also section 4.1.1. still discusses the problem as if there was an inherent bias depending on reward formulation. Furthermore, I already pointed out several problems and errors related to the illustrative example (e.g. analysing an intermediate state of the algorithm, rather than a fixed-point). Maybe you could prove for your code example that AIRL does not converge and show a plot that compares the averages trajectory length for the buggy implementation with my naive fix.
> > > > >
> > > > > Section 4.2. seems like the main technical contribution. The last paragraph still looks fishy to me and the reported problem of using the analytically derived return seems to result from an assumed infinite horizon formulation. I think that the MDP formulation used for handling absorbing states seems to assume (potentially very large) finite horizons and hence, R_T should at least theoretically depend on the current time step. Given that both equations are analytically equivalent, one equation can not be more stable than the other. When, however, the explicit summation is performed until a given horizon is reached, whereas the closed form solution assumes an infinite horizon, the returned values differ and the closed form solution is simply not sound.
> > > > >
> > > > > [1] Time Limits in Reinforcement Learning, Fabio Pardo, Arash Tavakoli, Vitaly Levdik, Petar Kormushev,
> > > > > Proceedings of the 35th International Conference on Machine Learning, PMLR 80:4045-4054, 2018.

---

> > > > > > ### Author Response · Authors · 2018-11-21
> > > > > > **Additional response to AnonReviewer2**
> > > > > >
> > > > > > Thank you for your detailed and encouraging response.
> > > > > >
> > > > > > We have updated the paper to try to address your suggestions. We hope that this revised version more appropriately positions the contribution and draws a clear distinction between MDP formulation and algorithm, as per your suggestion. In particular:
> > > > > >
> > > > > > 1. We now make it clear that the correct handling of absorbing states is something that should be applied to any inverse reinforcement learning or reward learning algorithm, whether adversarial or otherwise, and is independent of the DAC algorithm in that sense.
> > > > > >
> > > > > > 2. We have added the suggested citation and other papers that discuss time limits (Pardo et al: https://arxiv.org/abs/1712.00378, Tucker et al: https://arxiv.org/abs/1802.10031 ) in the related work section.
> > > > > >
> > > > > > 3. In Section 3, we've added a discussion of time limits in MDPs, as well as a discussion of how temporal difference methods can handle infinite-horizon tasks with finite-horizon rollouts (which is what DAC does also). Please see the last paragraph of the section.
> > > > > >
> > > > > > 4. As per your suggestion, we have removed the illustrative example in Section 4.1.1. While we do believe that an example would help illustrate the issue to the reader, we understand your reservations against the illustrative example. We would like to attempt to add a better illustrative example in the final version(we just have not had time to do so), but we will be sure to make an additional post about it if we do, to confirm that it is satisfactory.
> > > > > >
> > > > > > 5. For the sake of clarity, we removed the last paragraph from section 4.2 that discusses our choice of the implementation of bootstrapping for the terminal states.
> > > > > >
> > > > > > We appreciate your patience, and would appreciate it if you took another look at the paper and let us know if this has addressed your concerns.

---

> > > > > > > ### Comment · AnonReviewer2 · 2018-11-21
> > > > > > > **Revised version is much better**
> > > > > > >
> > > > > > > Thanks for the quick revision. The submission is much better now.
> > > > > > > I updated my review to take the revised version into account, however, I did not feel comfortable adapting my rating quite yet (please refer to the review for an explanation).
> > > > > > > I encourage you to further revise the submission.

---

> ### Author Response · Authors · 2018-11-22
> **Response to the update from AnonReviewer2**
>
> We again would like to emphasize that we appreciate your patience and valuable feedback that helps us to improve our submission.
>
> We have updated the paper to try to address your suggestions. In particular:
>
> 1) As per your suggestion, we extended the last paragraph of Section 3.1 in order to clarify our discussion on episode termination because of time limits. We believe that it adds clarity to the paper because it discusses  termination  states in more detail. It also  explains the difference between absorbing states and rollout breaks. For a detailed discussion on implementation specific biases in algorithms (GAIL/AIRL) please refer section 4.1.
>
> 2) In Section 4.1, we enumerate  papers affected by this problem, with specific instances. For each paper that we cite in this section, we consider the official implementations provided by the authors. In the same section, we further elaborate on how exactly these algorithms  are affected by the issue.
>
> 3) In section 4.2, we assume infinite horizon for R_T since the series converges ( assuming reward bounded by r_max, the series is bounded by gamma/(1-gamma) r_max and thus can be computed either analytically or will converge in the limit using TD updates, section 3.1 now also includes a clarification of this point). We also extended Section 4.2 to clarify how absorbing states can be used by the AIL algorithms and how the corresponding transitions affect estimations of returns. Please see the second paragraph of Section 4.2.
>
> 4) Regarding revisiting the illustrative example, we agree that the same reasoning might apply to Inverse RL algorithms in general and we appreciate your suggestion regarding the analysis of this simple example for MaxEnt-IRL. We unfortunately will not be able to add such an experiment before the end of the revision period, but we have added some discussion in Section 4 that the basic principle applies also to other IRL methods. This can be considered as an interesting direction of future work. e will attempt to add a better illustrative example in the final version (we just have not had time to do so), and will make sure to update the reviewers about it.
>
> 5) We fixed the wrongly referenced section. Thanks for catching this.
>
> We hope that this revision of our submission will address your concerns.

---

### Official Review · AnonReviewer1 · 2018-11-05
**A Review on Adversarial Inference by Matching Priors and Conditionals Discriminator-Actor-Critic: Addressing Sample Inefficiency and Reward Bias in Adversarial Imitation Learning**

**Rating:** 8
**Confidence:** 2

**Review:**

The authors find 2 issues with Adversarial Imitation Learning-style algorithms: I) implicit bias in the reward functions and II) despite abilities of coping with little data, high interaction with the environment is required. The authors suggest "Discriminator-Actor-Critic" - an off-policy Reinforcement Learning reducing complexity up to 10 and being unbiased, hence very flexible.

Several standard tasks, a robotic, and a VR task are used to show-case the effectiveness by a working implementation in TensorFlow Eager.

The paper is well written, and there is practically no criticism.

---

> ### Public Comment · (anonymous) · 2018-11-06
> **Question on expertise of the reviewer in this domain**
>
> It doesn't seem that the reviewer has put any efforts in appreciating or criticising the paper and has merely summarised the paper in a few lines.
> Please provide proper analysis for your acceptance decision and rating

---

> ### Author Response · Authors · 2018-11-14
> **Response to AnonReviewer1**
>
> We thank the reviewer for the feedback and appreciate the strong recommendation.

---

### Public Comment · (anonymous) · 2018-10-24
**Some comments on the persuasion and sufficiency of experiments**

I found there is a significant gap between the performances of GAIL reported by the authors and stated in the original GAIL paper(https://papers.nips.cc/paper/6391-generative-adversarial-imitation-learning-supplemental.zip). Since the authors emphasized that they use the original implementation(https://www.github.com/openai/imitation), such empirical results could be doubtful. Can the authors comment on that?

Another comment is about the sufficiency on experiments. Since DAC is a combination of an improved adversarial reward learning mechanism and off-policy training,  evaluations on ablations are needed to clarify which part actually accounts for the improvement on performance or training efficiency. Moreover, I think GAIL with off-policy training should also be a baseline to further validate that whether the unbiased reward learning introduced by the authors could eliminate the sub-optimality.

---

> ### Author Response · Authors · 2018-10-26
> **Original GAIL results and ablation experiments**
>
> Thank you for your comments.
>
> At the moment, we plot results only for 1 million steps. In the original implementation of GAIL, the authors use 25M steps to report the results. With 25M steps we are able to replicate results reported in the original GAIL paper. We do have one example of how the methods compare to each other when trained for 25M steps in our submission. This can be seen in the top left sub-plot in Figure 4. We will add the plots with 25M steps in the next update of the paper.
>
> We perform ablation experiments and visualize the results in Figure 6. The ‘no absorbing’ baseline corresponds to off-policy GAIL while the red line corresponds to DAC. Thanks for pointing this out. We will add a clarification in the text to make the comparison clearer.

---

> > ### Public Comment · (anonymous) · 2018-10-29
> > **Response**
> >
> > I appreciate the authors' response to the comments, and it did address some of my concerns. However, I still have some questions:
> >
> > 1. Could the authors provide the comparisons among DAC, GAIL w/ PPO, and GAIL w/ TRPO for 25M steps for all the 5 tasks (in Fig.4)?
> >
> > 2. Why the authors only evaluate such no absorbing experiments on KUKA tasks? Could the authors provide the results of this baseline on the 5 tasks used in Fig.4?

---

> > > ### Author Response · Authors · 2018-10-30
> > > **RE: Response**
> > >
> > > Thank you again for your comments.
> > >
> > > 1. We did not have sufficient time to collate these results before the deadline, but we will add them to the appendix for a future revision.
> > >
> > > 2. In Fig. 7, we run the absorbing state versus non-absorbing state experiments on the more standard Hopper and Walker2D environments. We understand those experiments are with AIRL algorithm and it will be more comprehensive if we ran the same experiment with GAIL algorithm and environments from Fig. 4. However, we were constrained by the page limits and chose to show how our fix to the reward bias not only works across different adversarial algorithms (GAIL in Fig. 6 and AIRL in Fig. 7) but also works on demonstrations collected from humans on a Kuka arm. We will add the figures for the experiments you mentioned in the comment to the next version of the paper.

---

### Public Comment · (anonymous) · 2018-10-28
**Another paper with off policy imitation learning**

There is another paper which has also combined off-policy training with imitation learning.
The only significant contribution of this paper then seems to be unbiased rewards.
I think the authors should provide more rigorous analysis of what exact effects the absorbing state introduces.
https://arxiv.org/pdf/1809.02064.pdf

---

> ### Author Response · Authors · 2018-10-30
> **RE: Another paper with off policy imitation learning**
>
> Thank you for sharing the link. The arxiv paper linked is concurrent work. As such our off-policy algorithm was novel at time of release and remains a primary contribution of this work. We will add this paper to the related work section as a concurrent work in the next update.
>
> The requested ablation study is already presented in Fig. 6 and Fig.7 where we compare adversarial imitation learning approaches with and without the absorbing states. Due to the bias present in the original reward, the baseline without absorbing state information fails to learn a good policy. We derive why this happens in Section 4.1.
>
> Also, we would like to emphasize that our paper is not limited to off-policy training but also addresses other issues of adversarial imitation learning algorithms. We first identify the problem of biased rewards, which we then experimentally validate across GAIL and AIRL (note that the other paper is centered around GAIL, and not adversarial imitation learning in general). Following that we introduce absorbing states as a fix for this issue, while empirically validating that our proposed solution solves tasks which are unsolvable by AIRL.

---

### Public Comment · (anonymous) · 2018-11-03
**Practical Details for Reproducing Results**

I enjoyed reading your submission and was trying to reproduce some of your results. I am using Soft Actor-Critic with somewhat different model sizes than yours and had some practical questions that could help me be more effective. I was wondering:

- What is the batch size you use for the updates? How large is your replay buffer size (do you store all previous trajectories)?
- In your algorithm box, in the update section, it says "for i = 1, ..., |\tau|". Does this mean that for example for halfcheetah environment you do 1000 updates every time you generate a trajectory?
- Would you be able to also include the numeric reward value your experts achieve on the tasks?
- Could you elaborate on the specific form of gradient penalty you use and the coefficient of the gradient penalty term?

And, one separate question: Have you also tried simultaneously updating the discriminator and policy instead of the alternating scheme shown in the algorithm box?

Thank you!

---

> ### Author Response · Authors · 2018-11-05
> **Response**
>
> Thank you for your comments!
>
> Since our algorithm uses TD3 (https://arxiv.org/pdf/1802.09477.pdf), we highly recommend to use the original implementation of the algorithm (https://github.com/sfujim/TD3). Our reimplementation of TD3 reproduces the results reported in the original paper. Reproducing results with SAC might be harder since SAC requires tuning a temperature hyperparameter that might require additional efforts in combination with reward learning.
>
> 1) We used the batch size equal to 100. We kept all transitions in the replay buffer.
> 2) That’s correct. For HalfCheetah, after performing 1K updates of the discriminator we performed 1K updates of TD3. During early stage of development we tried the aforementioned suggestion of simultaneously updating the discriminator and policy, and it produced worse results.
> 3) Yes, we will include it in  the appendix.
> 4) We used gradient penalty described in https://arxiv.org/abs/1704.00028 and implemented in TensorFlow https://www.tensorflow.org/api_docs/python/tf/contrib/gan/losses/wargs/wasserstein_gradient_penalty with a coefficient equal to 10.
>
> An additional note regarding reproducing results. Please take into account, that depending on when you subsample trajectories to match the original GAIL setup, you need to use importance weights. Specifically, if you first subsample expert trajectories taking every Nth transition, and then add absorbing states to the subsampled trajectories, you will need to use importance weight 1/N for the expert absorbing states while training the discriminator. We will explicitly mention this detail in next version of the submission.
>
> We would like to emphasize that upon publishing the paper we are going to open source our implementation.
>
> Feel free to request any additional information. We will be glad to provide everything to help you to reproduce our results.

---

### Public Comment · (anonymous) · 2018-11-08
**Question about Details of Algorithm and ablation study**

I enjoyed reading your submission, and I am now trying to add absorbing state to AIRL.
I have 3 questions. First and second questions are about how to learn Q_theta(s_a,・) and third is about ablation study.

Three questions are below.

1.  I think that the target of Q_theta(s_a, ・) is logD(s_a,・)-log(1-D(s_a,・)) +  γQ_theta(s_a,・). Is this right?

2. What did you use as action at absorbing states for calculating D(s_a,・) or Q_theta(s_a,・)? You use random value?

3. Did you investigate the effect of only absorbing states on on-policy GAIL or AIRL ? Did GAIL+absorbing states or AIRL + absorbing states work better than GAIL or AIRL?

Thank you!!

---

> ### Author Response · Authors · 2018-11-09
> **Response**
>
> Thank you for your comments.
>
> 1. Yes, that’s correct (using TD3 algorithm). For the target part it’s s’ and action is produced by the action target network:  ||logD(s_a,・)-log(1-D(s_a,・)) +  γQ_theta_target(s’, A_target(s’),・)　-Q_theta(s, a,・) ||**2.
> 2. We used zero actions for the absorbing states.
> 3. No, we investigated it only with off-policy case. For the off-policy version of your second question, see Figures 6 and 7. However, the part related to absorbing states is independent of off-policy training.

---

### Public Comment · ~Sheldon_Benard2 · 2019-01-08
**ICLR 2019 Reproducibility Challenge Description**

As part of the ICLR 2019 Reproducibility Challenge, we (Sheldon Benard, Vincent Luczkow, & Samin Yeasar Arnob) attempted to replicate the results of Discriminator-Actor-Critic:  Addressing Sample Inefficiency and Reward Bias in Inverse Reinforcement Learning.  Discriminator-Actor-Critic (DAC) is an adversarial imitation learning algorithm. It uses an off-policy reinforcement learning algorithm to improve upon the sampling efficiency of existing methods, and it extends the learning environment with absorbing states and uses a new reward function in order to achieve unbiased rewards.  We were able to achieve comparable rewards and sample efficiency on two of the four environments.  For the environments in which we were unable to reproduce the original results, we will continue to perform experiments and converse with the authors.   All  of  our  code  is  available  at:

https://github.com/vluzko/dac-iclr-reproducibility

---

> ### Public Comment · ~Samin_Yeasar_Arnob1 · 2019-02-23
> **Update on Reproducibility**
>
> Hi, even though we couldn't replicate the results in time for reproducibility challenge, after consulting with the authors we're able to get the similar results (we did our experiments on Hopper-v2, Ant-v2, Half-cheetah-v2, Walker2d-v2) with occasional performance drop (except for Half-cheetah-v2). According to the authors they used 10 random seeds and averaged over the found results but due to limited computational resource we're not able to confirm that we do confirm that it replicates the performance and sample efficiency from figure 4. I really appreciate authors effort for taking the time to evaluate our work and getting back to us.

---

> > ### Author Response · Authors · 2019-02-28
> > **Our implementation**
> >
> > Thanks for your feedback regarding our paper! We extremely appreciate your efforts.
> >
> > We have open sourced the original implementation of our paper:
> > https://github.com/google-research/google-research/tree/master/dac

---

### Meta-Review · Area_Chair1 · 2018-12-18
**Well written paper highlighting and fixing a common problem in Adversarial Imitation Learning algorithms**

**Confidence:** 4
**Recommendation:** Accept (Poster)

**Metareview:**

This work highlights the problem of biased rewards present in common adversarial imitation learning implementations, and proposes adding absorbing states to to fix the issue. This is combined with an off-policy training algorithm, yielding significantly improved sample efficiency, whose benefits are convincingly shown empirically. The paper is well written and clearly presents the contributions. Questions were satisfactorily answered during discussion, and resulted in an improved submission, a paper that all reviewers now agree is worth presenting at ICLR.